# Impact of Harvest Month and Drying Process on the Nutritional and Bioactive Properties of Wild *Palmaria palmata* from Atlantic Canada

**DOI:** 10.3390/md21070392

**Published:** 2023-07-03

**Authors:** Bétina Lafeuille, Éric Tamigneaux, Karine Berger, Véronique Provencher, Lucie Beaulieu

**Affiliations:** 1Département de Science des Aliments, Faculté des Sciences de l’Agriculture et de l’alimentation (FSAA), Université Laval, Québec, QC G1V 0A6, Canada; betina.lafeuille.1@ulaval.ca; 2Institut sur la Nutrition et les Aliments Fonctionnels (INAF), Québec, QC G1V 0A6, Canada; etamigneaux@cegepgim.ca (É.T.); veronique.provencher@fsaa.ulaval.ca (V.P.); 3Centre Nutrition, Santé et Société (NUTRISS), Université Laval, Québec, QC G1V 0A6, Canada; 4École des Pêches et de L’aquaculture du Québec, Cégep de la Gaspésie et des Îles, Québec, QC G0C 1V0, Canada; 5Merinov, Grande-Rivière, QC G0C 1V0, Canada; karine.berger@merinov.ca; 6École de Nutrition, Faculté des Sciences de l’Agriculture et de l’Alimentation (FSAA), Université Laval, Québec, QC G1V 0A6, Canada; 7Québec-Océan, Université Laval, Québec QC G1V 0A6, Canada

**Keywords:** red macroalgae, wild, dried, antioxidant activity, ACE inhibitory activity

## Abstract

The macroalga *Palmaria palmata* could be a sustainable and nutritional food resource. However, its composition may vary according to its environment and to processing methods used. To investigate these variations, wild *P. palmata* from Quebec were harvested in October 2019 and June 2020, and dried (40 °C, ≃5 h) or stored as frozen controls (−80 °C). The chemical (lipids, proteins, ash, carbohydrates, fibers), mineral (I, K, Na, Ca, Mg, Fe), potential bioactive compound (carotenoids, polyphenols, β-carotene, α-tocopherol) compositions, and the in vitro antioxidant activity and angiotensin-converting enzyme (ACE) inhibition potential of water-soluble extracts were determined. The results suggested a more favorable macroalgae composition in June with a higher content of most nutrients, minerals, and bioactive compounds. October specimens were richer only in carbohydrates and carotenoids. No significant differences in antioxidant or anti-ACE inhibitory activities were found between the two harvest months. The drying process did not significantly impact the chemical and mineral compositions, resulting in only small variations. However, drying had negative impacts on polyphenols and anti-ACE activities in June, and on carotenoids in October. In addition, a concentration effect was observed for carotenoids, β-carotene and α-tocopherol in June. To provide macroalgae of the highest nutritional quality, the drying process for June specimens should be selected.

## 1. Introduction

In recent years, with the exponential increase in the world population and the perspective of sustainability, new food sources have been studied [1]. Taking into account economic, social, and environmental components [1,2], more sustainable food alternatives are being considered, and among them, seaweeds have become increasingly popular in Western Countries [3,4].

Seaweeds have been an integral part of Asian culinary culture for a very long time. Asia is the leading producer, with China accounting for 59.5% of global algal farming in 2020 [5]. In Europe and North America, macroalgae consumption is more limited to certain areas, such as Ireland, Scotland, Brittany (France), Nova Scotia (Canada) or Maine (USA) [6,7]. In Canada, some coastal Indigenous groups and Irish immigration in the 19th century contributed to traditional local seaweed consumption [6]. In the Western world, macroalgae are mainly used in the food industry as an ingredient, in pharmaceutical and cosmetic production, or in animal feed [3,7], but they are still an uncommon part of most Canadians’ eating habits [6]. The ever-increasing commercial availability of seaweed food, however, could favor future positive changes [8].

Macroalgae are a large family of marine plants divided into several groups. The Ochrophyta are brown and include 1500–2000 species, the Chlorophyta are green with 8000 species, and the Rhodophyta are red with 6000 to 7000 species worldwide [9,10]. One of the most common red seaweeds in the North Atlantic Ocean is *Palmaria palmata* (*P. palmata*), also called dulse [3]. The macroalga *P. palmata* is found in sheltered and moderately exposed areas of the intertidal and subtidal zones (maximal depth of 20 m). It lives on hard substrates, like rocky shores, but can also be found as an epiphyte on the stipes of brown seaweeds such as *Laminaria hyperborea*, *Laminaria digitata*, or *Saccharina latissima*. It is a pseudo-perennial small species whose frond length usually varies from 10 to 20 cm to a maximum of 50–70 cm, and it can regrow new fronds every year [3,11,12,13]. Red algae in general, and especially *P. palmata*, are known to be a valuable source of proteins, with protein contents ranging from 7 to 19% dry weight (DW) with approximately 30% of the essential amino acids counted in the total amino acid fraction [12]. In addition, they also contain high proportions of polysaccharides (38–74%) such as carrageenan, and a large variety of minerals (12–37%) such as I, K, Na, Ca, Mg, or Fe [7,14,15]. Despite the low proportion of lipids (0.2–3.8%), *P. palmata* can contain liposoluble vitamins such as β-carotene (provitamin A) or tocopherol (vitamin E) and omega-3 [7,16,17]. In addition to their nutritional benefits, they contain bioactive compounds responsible for different bioactivities. Peptides derived from proteins and proteins involved in photosynthesis have demonstrated antioxidant and ACE (angiotensin-converting enzyme) inhibition activities [18,19]. Immunomodulatory, anticoagulant, or antihyperlipidemic effects associated with the presence of carrageenan have also been detected with other red macroalgae [15,20]. Carotenoids such as lutein or β-carotene [16,17] and polyphenols have demonstrated antioxidant properties [16,21]. Macroalgae composition and bioactivities could vary according to the growing environment, the season, the light intensity, or the availability of nutrients [3,16,18,22,23,24,25]. Wild *P. palmata*, in the North Atlantic regions have a seasonal life cycle with a period of reproduction and growth during winter and spring, and no or negative growth during summer and autumn [3,11,12,25,26]. Changes in the composition of certain chemicals and bioactives therefore seem linked to the concentration of photosynthetic pigment, whose synthesis is favored during winter and spring due to environmental factors such as light irradiance and seawater nutrient concentrations [24,25]. This strong seasonality has been investigated in many studies [3,13,18,23,24,25,27,28] and some research has been carried out in Canada. This includes the work of Vasconcelos et al. [29], which highlighted variations in the chemical composition of Quebec macroalgae over several months. Indeed, *P. palmata* harvested in Île-de-la-Madeleine (QC, Canada) had higher protein, mineral, and lipid contents during the summer (June and July), whereas carbohydrate content was high in the autumn (October). More studies are needed in this area, especially in the St. Lawrence estuary, which is an important natural cradle for the wild biomass of macroalgae.

Once macroalgae are harvested, food processing is necessary to stabilize the product over time as it contains more than 75 to 85% water [30,31]. Moreover, in the Western world, because crude macroalgae are not really integrated into food habits, processing methods such as drying, blanching, freezing, etc., could favor consumption [8]. However, different food processing methods may induce variations in macroalgae composition and bioactivity. Overall, two major principles are applied to seaweed preservation: heat and cold. The oldest and most widespread processing method is drying, which consists of removing water, and reducing degradation and the growth of microorganisms [30,31]. One of the most ancient drying methods is sun drying, which is the most accessible method, but requires a warm and sunny climate and does not allow the constant application of a precise temperature [6,32]. Other drying methods, such as oven drying, compensate for these disadvantages and allow precise control of the applied heat [30,31]. The impact of drying on seaweed composition seems to be mainly related to the temperature used. This could induce changes in the nutritional value or bioactive potential with losses of heat-sensitive molecules such as polyphenols or carotenoids [31,33,34]. A previous study using another edible red seaweed, *Pyropia orbicularis*, found a decrease in pigment content and antioxidant activity when using a drying method [35]. The second common treatment principle is the application of low temperature (freezing), which limits or even stops microbial metabolism and enzymatic activities [36]. The most efficient preservation method for long-term storage is freezing, corresponding to a storage temperature under the food freezing point, leading to ice crystal formation [36,37,38]. This phenomenon can cause changes in membrane integrity, leading to losses of nutritional compounds such as amino acids during thawing [39]. Trials performed on *P. palmata* frozen at −20 °C showed a loss of over 40% of the amino acid content compared to macroalgae held in seawater at 5 °C [39]. The overall effect of freezing on macroalgae composition is not completely clear but seems to be more efficient than drying for preserving bioactive compounds and bioactivities, such as antioxidant potential. A study using *Phyllariopsis brevipes* (formerly *Phyllaria reniformis*) found a higher antioxidant activity for frozen (−18 °C) compared to dried (38 °C) seaweed [40]. In contrast, previous studies using *S. latissima* have found a similar nutritional composition and antioxidant potential (depending on the harvest month) between frozen (−80 °C) and dried (40 °C) macroalgae, whereas some losses were detected for carotenoids and polyphenols in dried (40 °C) specimens [41]. Studies on *P. palmata* need to be expanded in order to understand the impact of the month of harvest and food processing method on its quality.

As mentioned above, among the many bioactive compounds in seaweeds are proteins and peptides [18,19]. In order to test their bioactive capacities, the extraction of these compounds is necessary and can be carried out according to the different aqueous methods referenced in the literature [9,19,27,42,43,44]. In addition, for the bioactivity measurements, many protocols exist, as is the case for antioxidant capacity tests. Indeed, several tests are used to measure the antioxidant capacity based on two principles: the electron transfer reaction (such as the ferric reducing antioxidant capacity assay [FRAP] or the 2,2-diphenyl-1-picrylhydrazyl [DPPH] test) or the hydrogen atom transfer (such as oxygen radical absorbance capacity assay [ORAC] or total radical trapping antioxidant parameter test [TRAP]) [45,46]. Since the use of a single technique is not recommended [47], previous studies have used at least two tests [19,27,42,43,44]. In general, however, this wide variety of extraction protocols and bioactive tests makes comparisons between studies difficult and should be considered carefully.

The objectives of the present study aimed to determine the impact of (1) the harvesting month and (2) the drying process on chemical (lipids, proteins, ash, carbohydrates, fibers), mineral (I, K, Na, Ca, Mg, Fe) and bioactive compound composition (carotenoids, polyphenols, tocopherol, β-carotene), as well as in vitro bioactivity potential (antioxidant and ACE inhibitory) on wild *P. palmata* harvested on the shores of Eastern Canada.

## 2. Results and Discussion

### 2.1. Chemical Composition

#### 2.1.1. Crude *Palmaria palmata*

The chemical composition of crude *P. palmata* is shown on a dry matter basis in Table 1. Overall, crude *P. palmata* harvested in June was richer in most compounds, except for carbohydrates, which were more concentrated in October. Lipids, proteins, and minerals were significantly higher in samples harvested in June (4.64 ± 0.19%, 16.19 ± 0.09% and 27.26 ± 0.29% dry weight (DW), respectively) than in October (1.17 ± 0.18%, 11.23 ± 0.17% and 12.30 ± 0.04% DW, respectively). Carbohydrates were significantly higher in specimens harvested in October, with concentrations varying from 75.10% to 75.50% DW, whereas their concentration was significantly lower in June (51.91 ± 0.57% DW). On the other hand, fiber content was significantly higher for macroalgae harvested in June (36.88 ± 0.10% DW) compared to those harvested in October (35.81 ± 0.27% DW).

The chemical composition of October and June seaweeds were fairly consistent with the literature. The reported proportions of protein in *P. palmata* varied from 7 to 19% DW, ash varied from 12 to 37% DW, lipids varied from 0.2 to 3.8% DW, and carbohydrates varied from 38 to 74% DW [7,12]. Furthermore, the observed chemical compositions echo the life cycle of Atlantic *P. palmata*, characterized by a reproductive period and nutrient storage in winter, growth in spring, end of growth and senescence in summer, and growth slowdown period in autumn [3,11,25,26]. Thus, the strong seasonality of seaweed due to environmental variations (such as light or nutrient availability) has already been described in the literature. In fact, previous studies have shown that the highest protein content was detected in winter and early spring, when seawater is particularly rich in nutrients, and the lowest during summer and autumn [24,25]. Even though June seaweeds in this study appeared to be the most protein-rich specimens compared to October seaweeds, previous observations of *P. palmata* in France and Norway have shown that winter specimens can reach approximately 25% DW [3]. Seasonal protein synthesis could be related to the *P. palmata* pigment study of Schmid et al. [25], which demonstrated that photosynthetic pigment concentrations were dependent on the availability of light and nutrients from seawater. Thus, winter and early spring were characterized by low irradiance and a high nutrient seawater concentration that favor pigment synthesis, resulting in highly active photosynthesis and nutrient storage for later months. On the other hand, high temperature, high irradiance, and a lack of nutrients in the summer and autumn led to loss of photosynthetic efficiency and the use of nutrient reserves [3,25]. In the macroalgae harvest area of the present study, nutrient stratification in the water column typically occurs from late spring to late September and reduces the availability of nutrients (such as nitrogen) [48]. Thus, the observed differences in protein levels could be related to the fact that June specimens have not consumed their reserves, while October macroalgae have used their reserves during the summer and have not had time to replenish them. Lipid content was also investigated and correlated to that nutrient cycle because of their role in photosystems [25]. In general, it was not surprising to observe a higher level of most chemical compounds in June compared to October because of the reserves in late spring, which were then likely consumed during the summer. For carbohydrates, it is interesting to note that whatever the month of harvest, both macroalgae had similar proportions of fiber, while the total carbohydrate content was usually higher in October specimens. This observation suggests that the difference in total carbohydrate content could be due to higher proportions of monosaccharides such as floridosides. Floridoside is the main carbon reserve synthesized in *P. palmata* due to high light exposure; it is accumulated during summer and is important for vegetative propagation during winter [26]. Its synthesis seems to have taken place after the month of June and seems to constitute the detectable reserves in October. These results demonstrated an inverse correlation between protein/ash and carbohydrate contents, which has previously been reported in the literature [49]. Moreover, the study of Vasconcelos et al. [29] found similar variations in the chemical composition of *P. palmata* from Île-de-la-Madeleine (Gulf of Saint-Lawrence, QC, Canada) harvested in June, July, and October, 2015. Their results showed an increase in carbohydrates and a decrease in protein and ash during summer and autumn, and similar proportions of fiber between July and October.

#### 2.1.2. Dried *Palmaria palmata*

The chemical composition of dried *P. palmata* is presented on a dry matter basis in Table 1. First, interactions between the month of harvest and drying treatment (*p*-value < 0.05) were observed, meaning that the impact of processing on macronutrient content was variable between the different months. Similar to crude *P. palmata*, dried samples harvested in June were significantly richer in lipids, proteins, ash, and fiber (3.65 ± 0.21%, 17.32 ± 0.17%, 26.18 ± 0.05% and 37.80 ± 0.22% DW, respectively), and dried October specimens were richer in carbohydrates (74.94 ± 0.13% DW). In comparison to crude *P. palmata* harvested in June, the results presented above for lipids and ash were slightly but significantly lower. In contrast, proteins and fiber were higher. No significant difference in carbohydrates was found between crude and processed seaweeds.

The results obtained for chemical composition were consistent with previous studies on dried Quebec *P. palmata*. Indeed, Lafeuille et al. [50] reported, on specimens harvested from the same site (Forillon, QC, Canada), an average lipid content of 0.41% DW, protein 16.05% DW, ash 20.48% DW, carbohydrate 63.05% DW, and fiber 35.26% DW. Furthermore, the Vasconcelos et al. [29] study on dried *P. palmata* harvested from six different locations in the Gulf of Saint-Lawrence (Rivière-au-Renard, Gaspé, Grande-Rivière, Pabos, Newport, Sept-Îles and Île-de-la-Madeleine) during 2015 found, in terms of dry weight, less than 2.1% was due to lipids, 9.0 to 16.9% to proteins, 11.1 to 40.0% to ash, 34.6 to 63.2% to carbohydrates, and 7.4 to 12.8% to fibers, similar to our results. Overall, the month of harvest (and by extension, the growth environment and the life cycle phase) has the largest effect on the chemical composition of *P. palmata*, with the drying method having only a small effect. The principle of the drying method is just the removal of water, and is not known to induce large losses of macronutrients [31]. However, environmental conditions, as discussed above (Section 2.1.1), had a tremendous impact and almost entirely determined the chemical compositions of dried *P. palmata*.

### 2.2. Mineral Composition

#### 2.2.1. Crude *Palmaria palmata*

The mineral composition of crude *P. palmata* is shown on a dry matter basis in Table 2. Mineral analysis revealed that the highest amounts of sodium (Na) (2.58 ± 0.06 g/100 g), calcium (Ca) (317.00 ± 21.00 mg/100 g), magnesium (Mg) (313.00 ± 9.00 mg/100 g), and iron (Fe) (436.67 ± 35.00 mg/100 g) occurred in *P. palmata* harvested in June, and iodine (I) and potassium (K) occurred in similar proportions in the October and June harvests.

The proportions of Na, Mg, and I for both harvest months were close to the results reported previously in the literature. Indeed, Mabeau and Fleurence [51] reported a Na content of 1.7 to 2.5 g/100 g DW and Mg content of 170 to 500 mg/100 g DW, and Mouritsen et al. [32] reported an I content of 0.5 to 1.0 mg/100 g DW. Since both Na and Mg are involved in seaweed photosynthesis [52], spring seawater is known to be richer in nutrients [24,25], and the macroalgae showed growth phase signs in June, it was not surprising to find a higher content in the June specimens. Compared to other foods, Mg content in the seaweeds harvested in June and October was much higher than in spinach (a great source of Mg [53,54]), which has approximately 79 mg/100 g [55]. The amount of Mg in the June macroalgae was much closer to that of dark chocolate (70–85% cocoa, known to be rich in this mineral [56]), which has, on average, 228 mg Mg/100 g [57]. Brown macroalgae such as *S. latissima* are known to have a high I content [14,15], and previous studies on Quebec specimens have detected an I content 10–100 times higher in *S. latissima* [41] than in *P. palmata*. This difference could be considered an advantage that allows a higher consumption of these red macroalgae because the maximum recommended intake of I is approximately 1.1 mg per day for an adult [14,15]. The phenomenon behind the fact that no significant difference was found for the I content between June and October is not clear. Seawater is richer in nutrients in spring than in autumn, but since I is involved in the production of secondary metabolites that may play a role as an antioxidant [52,58], it should probably have been higher in June, which was not the case. The K content from the October harvest (3.43 ± 0.06 g/100 g DW) was lower than the results of Mabeau and Fleurence [51], where K proportions ranged from 7.0 to 9.0 g/100 g DW. It is interesting to note that, as a strong source of K [59], bananas contain on average 358 mg K/100 g [60], thus, approximately 25 times less K than the macroalgae harvested in June, in comparable portions. The amount of Ca detected in the macroalgae harvested in both months was also lower (560–1200 mg/100 g [51]). However, this was largely higher than watercress (known to be rich in this mineral [61]), which contains, on average, 81 mg Ca/100 g [60]. As for the Na and Mg content, these were probably higher in June because of nutrient richness in spring and the growth phase of the seaweed. The Fe content in October seaweeds (30.00 ± 1.00 mg/100 g DW) agreed with the results of Mabeau and Fleurence [51] (from 15 to 140 mg/100 g), whereas in June, the Fe content was approximately three times higher than the maximum found in this previous study. In comparison with a good vegetable source such as raw parsley [62], which contains, on average, 6.20 mg Fe/100 g, the amount of Fe in *P. palmata* specimens harvested in June was higher [63]. The accumulation of Fe in *P. palmata* could be related to the synthesis of ferredoxin, which is an essential iron-related protein involved in photosynthesis that appears to be more active in spring than in summer and autumn [64]. The Na/K ratio of diets is a risk factor for the development of hypertension and cardiovascular disease, and according to World Health Organization recommendations, should be less than 0.6 [17,65]. In the present study, the Na/K ratios of wild *P. palmata* were usually under 0.6 and the lowest ratios were observed in June. For comparison, the Na/K ratio of raw shrimp is 5.00 [66]. The observed variations in the mineral composition of dulse within the harvest months and also reported in the literature were probably related to seasonal changes in the environmental parameters, such as the mineral load of seawater, which is higher in spring but lower during the summer and the autumn [23,24,67]. In addition, they were likely related to physiological changes during the life cycle of *P. palmata* [3,11,25,26]. In addition, deterioration of fronds during summer was observed when the fronds were frequently covered with epiphytes, leading to *P. palmata* having a less favorable overall composition at the end of summer [3].

#### 2.2.2. Dried *Palmaria palmata*

The mineral composition of dried *P. palmata* is presented in Table 2. Statistical analysis revealed the same trends as mineral composition in crude seaweed harvested over the two months. The levels of Na (2.20 ± 0.04 g/100 g), Ca (256.00 ± 2.00 mg/100 g), Mg (261.00 ± 4.00 mg/100 g), and Fe (352.33 ± 4.16 mg/100 g) in dried *P. palmata* were significantly higher in June than in October. The levels of I and K were similar in October and June. Thus, interactions between the harvest months and drying method (*p*-value < 0.05) were detected for the minerals investigated. In comparison to crude *P. palmata* harvested in June, the amounts of Na, Ca, Mg, and Fe in dried dulse were significantly lower. On the other hand, for specimens harvested in October, no significant difference in the mineral composition was found between crude and dried seaweeds.

The range of mineral proportions in dried *P. Palmata* from October and June were consistent with the results reported in previous studies. Maehre et al.’s [68] study on freeze-dried Norwegian *P. palmata* harvested in spring reported 360 mg Ca/100 g DW, 530 mg Mg/100 g DW, and 0.026 g I/100 g DW. Only Fe was mostly lower (10 mg/100 g DW) compared to the results of our study, suggesting that Quebec spring seawater was richer in Fe. This phenomenon could be explained by the composition of the rocky bed of the Gaspé Peninsula (harvesting area), which is composed of red sandstone containing hematite, an iron oxide [69]. Another study performed with freeze-dried Danish *P. palmata* harvested in September detected approximately 4.11 g K/100 g, 160.00 mg Mg/100 g, 30.70 mg Fe/100 g, 0.32 g Na/100 g, and 933.00 mg Ca/100 g, with a Na/K ratio of 0.08 [17]. Compared to those results, Quebec *P. palmata* harvested in October contained similar amounts of K, Fe, and Mg, but approximately seven times less Ca and four times more Na. October and June dried specimens had a higher Na/K ratio compared to the results of Parjikolaei et al.’s [17] study, but still remained below 0.6 [17,65]. The detected loss of minerals in June macroalgae might be linked to the Maillard reaction or mineral binding protein denaturation [70].

### 2.3. Bioactive Compound Content

#### 2.3.1. Crude *Palmaria palmata*

The bioactive compound content of crude *P. palmata* is shown on a dry matter basis in Table 3. *P. palmata* harvested in October showed a significantly higher proportion of carotenoids (371.00 ± 11.00 μg/g), while polyphenols (2.06 ± 0.05 mg GA/g), α-tocopherol (22.67 ± 2.08 μg/g) and β-carotene (33.33 ± 1.53 μg/g) contents were higher in June.

In the literature, several carotenoids have been detected in red seaweeds, such as lutein, zeaxanthin, or asthaxanthin [16,71]. But since it is known to be one of the major carotenoids [72], only asthaxanthin concentrations were determined and presented as carotenoid concentrations in this study. To the authors’ knowledge, no literature is available on the astaxanthin content of *P. palmata*. However, studies on other pigments have shown that the synthesis of carotenoids in this species are favorably linked to nitrogen and light [17,28]. In addition, a study of the red macroalga *Porphyra umbilicalis* from the North Atlantic reported higher carotenoid concentrations in September and November than in July. This was explained by the fact that the low irradiance of autumn favored pigment synthesis [73] and this was probably the case for the October specimens in the present study. On the other hand, β-carotene contents were higher in June, probably due to the high nutrient content of the water at this time of the year [28]. However, the concentrations detected were lower in both months than in previous studies, which reported 420 µg β-carotene /g in summer and 37 µg β-carotene /g in winter in *P. palmata* [16]. Compared to the β-carotene contents in vegetables such as carrots and mango (79.7 and 31.0 µg/g, respectively, which usually contain a high amount of this vitamin [17]), *P. palmata* had a lower amount than carrot in both months, but an equal amount to mango in June and a lower amount in October [17]. The total phenolic content (TPC) for both months was lower than that previously reported in the literature, probably due to variation in environmental factors. Indeed, *P. palmata* harvested in Iceland had a TPC of about 5 mg GAE/g [21], and in New Brunswick, Canada, this was 10.3 mg GAE/g [74]. Another study conducted on *P. palmata* harvested in spring on the west and east coasts of Grand Manan Island (NB, Canada) with variations in UV exposure, found no difference in TPC. In fact, the TPC of the western-side seaweeds was detected at 12.8 mg GAE/g, and for eastern-side specimens it was 12.7 mg GAE/g, suggesting a low impact of UV exposure [13]. Thus, the difference detected between June and October TPC in that study should not be related to the greater light exposure during the summer but probably to other environmental conditions, such as nutrient availability or water temperature. To the authors’ knowledge, no studies have been conducted on the α-tocopherol (vitamin E) content of *P. palmata*, but since α-tocopherol is a phenolic compound [13], it was not surprising to observe a higher level in June. Previous studies performed on red seaweeds have related varying concentrations from 0.01 µg/g DW in Peruvian *Chondracanthus chamissoi* [75] to 0.18 µg/g DW in Danish *Chondrus chrispus* [76]. Thus, detected α-tocopherol amounts ranging from 9.05 to 22.67 µg/g DW qualify wild *P. palmata* as a true vitamin-E-rich red species. Compared to common α-tocopherol-rich foods such as spinach or salmon [77,78], which contain 20.3 µg/g DW and 35.5 µg/g DW of α-tocopherol, respectively [55,79], *P. palmata* could be considered an interesting marine plant source of vitamin E in comparable serving sizes. It is therefore important to note that the actual consumed servings of seaweeds (80 g wet or approximately 16 g dried [80]) often remain lower than for spinach or salmon. The higher vitamin content in June could be due to the more active metabolism of the growth phase.

#### 2.3.2. Dried Palmaria palmata

The content of bioactive compounds for dried *P. palmata* is presented in Table 3. As with chemical and mineral compositions, interactions between the months of harvest and drying method (*p*-value < 0.05) were observed for bioactive compounds, thus compounds in dried seaweeds varied depending on the month of harvest. Compared to October, the drying process of June *P. palmata* produced the highest concentrations of carotenoids, polyphenols, α-tocopherol, and β-carotene (535.00 μg/g, 1.89 mg GA/g, 50.00 μg/g, and 47.67 μg/g, respectively). Dried October seaweeds were not significantly different from crude samples for polyphenols, α-tocopherol, and β-carotene contents. However, they differed in carotenoids, which were affected by the treatment. On another hand, dried June specimens had the highest levels of carotenoids, α-tocopherol, and β-carotene, and the lowest polyphenol content. Limited literature is available regarding the impact of drying treatment on bioactive compounds present in *P. palmata*. However, it is known that carotenoids and polyphenols are highly sensitive to heat and could be negatively affected by the drying treatment [33,34]. This seems to have been the case for carotenoids in dried October seaweeds and polyphenols in dried June macroalgae, but this phenomenon was not generalized. The higher levels of carotenoids and vitamins in the dried June samples could be due to a concentration effect related to drying. In a previous study, Parjikolaei et al. [17] detected approximately 2 μg β-carotene /g, in freeze-dried *P. palmata*, which is approximately 3.5 to 23.5 times less than in the macroalgae from this study. However, the work of Machu et al. [81] on dried *P. palmata* flakes reported a TPC of 31.8 mg GAE/g, which was generally higher than in our study.

### 2.4. In Vitro Bioactive Potential

#### 2.4.1. Protein Contents and Extraction Yields of Soluble *Palmaria palmata* Extracts

Crude *Palmaria palmata*

Extraction of water-soluble extracts >1 kDa from crude *P. palmata*, presented in Table 4, showed no significant difference in extraction yields and protein extraction yields (PEY) between October and June specimens. However, the protein contents of the water-soluble extracts were significantly higher in June macroalgae (29.03 ± 4.30% DW). This was not surprising because of the higher protein contents found in crude *P. palmata* in June (16.19 ± 0.09% DW, presented in Table 1). Whereas the objective of the extraction was to recover macroalgae proteins and peptides, the low PEY (maximum about 12% DW) showed limited extraction, probably due to interactions with polysaccharides [24,82,83]. A previous study performed protein water extraction on *P. palmata* harvested in July and October, and reported significantly lower protein contents in July (1.47 ± 0.04% DW) than in October (2.97 ± 0.04% DW) [27]. These observations regarding the amount of protein and the seasonality differed from our results. Indeed, the protein contents in this study were higher in June than in October, which would seem consistent with seawater nutrient availability in the spring, allowing protein synthesis that could produce phycobiliproteins (water-soluble pigment proteins) [84]. Different protein extraction results could be due to environmental factors (Ireland versus Quebec), seaweed conditions (freeze-dried versus defrosted), extraction method (deionized water versus phosphate buffer) or protein quantification method (Lowry method versus Dumas method).

Dried *Palmaria palmata*

Extraction performance on dried *P. palmata* (Table 4) showed no significant difference in protein content (approximately 23% DW), extraction yield (from 2.19 to 5.58% DW), and PEY (about 8.5% DW) between the October and June samples. Thus, the drying treatment appears to favor the extraction of a similar protein content, although the protein content of the dried June samples was higher than in October. In comparison with the crude June seaweeds, the drying process did not improve or limit extraction performance. This could be related to the fact that the crude samples were frozen and that both treatments (freezing and drying) can affect the integrity of the cell wall [30,31] and thus resulting in the same extraction performance. However, in water-soluble extracts from dried October specimens, the extraction yield (3.44 ± 1.25% DW) was significantly lower than that of crude samples (8.40 ± 0.24% DW) but did not affect the protein extraction. Interactions between the months of harvest and applied treatment were thus once again detected. In the literature, the PEY on dried *P. palmata* has been reported as being quite variable. Hell et al. [43] reported a PEY of 3.30% DW, while Bondu et al. [19] found 11.40 ± 3.12% DW, and Wang et al. [21] reported over 35% DW. The results of this study were close to those obtained by Bondu et al. [19], and differences could be due to polysaccharide content, environmental factors, drying techniques, or extraction protocols.

#### 2.4.2. Antioxidant Capacity of Water-Soluble *Palmaria palmata* Extracts

Crude *Palmaria palmata* extracts

Extraction ORAC and FRAP assays were performed on water-soluble crude *P. palmata* extracts in order to evaluate the antioxidant potential. ORAC values detected at 1250 µg of extract/mL were, on average, 30.36 ± 14.69 µmol TE/g in October and 43.26 ± 14.31 µmol TE/g in June, with no significant differences. FRAP values for the lowest concentration extracts (500 µg/mL) were, on average, slightly higher in both months (77.07 ± 32.12 µmol TE/g in October and 81.74 ± 28.68 µmol TE/g in June), with no significant differences. The ORAC and FRAP tests produced different results but this might be related to the different measuring mechanisms. The ORAC test measures the potential peroxyl radical scavenging capacity while the FRAP assay detects the potential reducing capacity of the extracts [18]. Overall, the antioxidant potential for these extracts did not change with the seasons. This result differed from previously reported ORAC and FRAP values for *P. palmata* harvested in Ireland, where July water extracts had a significantly higher antioxidant potential than October extracts. Harnedy et al. [27] reported ORAC values of 1.41 µmol TE/g in October and 2.18 µmol TE/g in July, and FRAP values of 81.35 µmol TE/g in October and 296.45 µmol TE/g in July. Differences compared to previous studies could be due to the extraction process, which was not identical, or due to environmental factors. In general, the ORAC values obtained by these authors were more than ten times higher than our results, whereas the FRAP values were close in October but lower in June [27]. Water-soluble extracts from *P. palmata* contained peptides and proteins that had already been identified in the literature. Beaulieu et al. [18] have studied the antioxidant potential of protein hydrolysis from *P. palmata* and have determined that major peptides with bioactivity were related to RuBisCo, a plant metabolic enzyme involved in photosynthesis, or to photosynthetic pigments, such as allophycocyanins or phycocyanins. These proteins and pigments are intrinsically linked to the rate of photosynthesis and should be more abundant in seasons with nutrient-rich seawater and high irradiance, such as the spring. Our results did not corroborate these published results, but the observed differences could be due to the sizes of proteins and peptides, since enzymatic hydrolysis allowed for better antioxidant activity [18] than for inherent seaweed peptide and protein synthesis. Another study performed on wild *P. palmata* grown with different light exposures highlighted the fact that reducing activity, detected with Yen and Chen’s 1995 method, was higher, with greater luminosity [13]. Explanations for this phenomenon were that the increase in UV radiation favored the synthesis of antioxidants with reducing capacity to help protect against photooxidative stress. In addition to proteins and peptides, water-soluble extracts also contained polyphenols and polysaccharides [43]. As described and discussed in Section 2.3.1, the TPCs of crude *P. palmata* were only slightly higher in June, but remained below the previously reported concentrations and did not substantially influence the measured antioxidant potential. This finding was the same for fiber (Section 2.1.1).

Dried *Palmaria palmata* Extracts

As was observed for water-soluble extracts from crude samples, water-soluble extracts from dried *P. palmata* did not show significant differences in ORAC and FRAP values between October and June, and were similar to crude/control samples. The ORAC values obtained from dried October specimens were, on average, 56.36 ± 16.46 µmol TE/g, and for dried June macroalgae, were 50.12 ± 19.92 µmol TE/g at the tested concentration of 1250 µg/mL. FRAP tests performed on dried macroalgae at 500 µg/mL produced higher values of 146.52 ± 65.64 µmol TE/g in October and 82.88 ± 15.31 µmol TE/g in June. Thus, the drying process did not impact the antioxidant capacity of *P. palmata*. Compared to previous studies on freeze-dried *P. palmata* water extracts, the values we obtained were close to the results of Wang et al. [21] who reported an ORAC value of approximately 35.8 µmol TE/g, and Harnedy and FitzGerald [42] who obtained an average of 45.17 ± 1.95 µmol TE/g. Using dried *P. palmata* (35 to 50 °C), Bondu et al. [19] reported ORAC values for the protein extract at the same concentration of less than 70 µmol TE/g, and Beaulieu et al. [44] found an antioxidant potential for the protein extract, <10 kDa, of less than 100 µmol TE/g. On the other hand, values were largely lower than Hell et al.’s [43] results for Quebec *P. palmata*, which was dried at 30 °C for 24 h, resulting in an ORAC value of 56.60 ± 5.06 mmol TE/g. The results of the FRAP tests seem highly dependent on the fraction size. Bondu et al. [19] reported FRAP values for chymotrypsin- and trypsin-hydrolyzed dried *P. palmata* extracts of 24.82 ± 1.91 µmol TE/g and 42.27 ± 3.42 µmol TE/g, respectively. Beaulieu et al. [44], however, detected a higher FRAP value on the unhydrolyzed protein extracts of dried *P. palmata* for the <10 kDa fraction (170.80 ± 15.21 µmol TE/g) than for the >10 kDa fraction (26.85 ± 0.23) at a concentration of 750 µg/mL. To the authors’ knowledge, no study has been conducted to investigate the effect of drying on the antioxidant activity of *P. palmata*. However, a study performed on another red seaweed, *Pyropia orbicularis*, found a reduction in antioxidant activity through sun drying (35 to 52.5 °C) and convective drying (70 °C, 120 min) compared to fresh specimens [35]. Thus, the ORAC value of methanolic extracts from fresh seaweeds was 22.5 µmol TE/g, while that of sun-dried and convective-dried samples was 9.38 µmol TE/g and 8.72 µmol TE/g, respectively. Drying techniques appeared to negatively affect photosynthetic pigments (such as phycocyanin) [35], which have previously been detected in *P. palmata* bioactive extracts [18], resulting in a loss of antioxidant activity. However, in the present study, the lack of a significant difference between dried and crude samples suggests a low impact of heat (40 °C) or the formation of new antioxidants such as melanoidins derived from the Maillard reaction, which may have taken place during drying [35]. As for the crude extracts, the PEY of the dried extracts were similar between October and June and were similar to the control samples, which could explain the lack of impact of the drying treatment. Polyphenols were significantly less present in dried June samples (Section 2.3.2), which did not impact the antioxidant potential of the dried macroalgae. Since water extracts may contain compounds other than peptides and proteins [43], further tests could help to evaluate their extraction as well as understand their actual impact on antioxidant potential.

#### 2.4.3. ACE-Inhibition Capacity of Water-Soluble *Palmaria palmata* Extracts

Crude *Palmaria palmata* Extracts

ACE-inhibitory percentages of crude *P. palmata* harvested in October and June are presented in Figure 1. Inhibition rates averaged 42.91% in the October extracts and 17.91% in June for both concentrations (8 and 10 mg of protein/mL corresponding to ≃38 mg of extract/mL and ≃ 47 mg of extract/mL, respectively). No significant difference was found between the two months (*p*-value > 0.05). A previous study performed on *P. palmata* harvested in July and in October in Ireland reported ACE IC_50_ values of >1.50 mg of extract/mL in July and >2.00 mg of extract/mL in October [27]. Therefore, 50% of ACE inhibition was reached by using very small amounts of extracts compared to our results. Another study performed on a different Irish red seaweed, *Porphyra dioica*, harvested in July and November, found ACE IC_50_ values relatively close to those obtained by the authors in [27], with 0.90 mg of extract/mL in July and 1.62 mg of extract/mL in November [85]. In addition, a significant difference was detected between these two months, with the November extracts showing higher ACE IC_50_ values. Overall, higher amounts of ACE-inhibitory compounds appear to be more available during the autumn. However, a previous study performed on frozen (−20 °C) Japanese *P. palmata* identified phycobiliproteins (phycoerythrin, phycocyanin, and allophycocyanin) as the original proteins of ACE-inhibitory peptides contained in water extracts [86]. Those proteins are pigment proteins that play an important role in macroalgae photosynthesis, and appear to be present at higher levels in winter and early spring, compared to summer and autumn [24,25], which is inconsistent with the previously published results. A higher anti-ACE effect was also obtained following the hydrolysis of proteins in water extracts [18,27]. Furuta et al. [86] showed ACE-inhibition rates went from approximately 30% for protein extracts to over 80% for hydrolyzed extracts [86]. Thus, the higher ACE-inhibition potential of crude *P. palmata* in autumn could be related to protein degradation, releasing bioactive anti-ACE peptides.

Dried *Palmaria palmata* Extracts

ACE-inhibitory percentages of dried *P. palmata* harvested in October and June are presented in Figure 1. Compared to crude specimens, ACE inhibition of dried *P. palmata* extracts was not significantly different. However, anti-ACE potential was significantly different between October and July. Therefore, the impact of the drying process differed across the seasons. Significant losses of inhibition rates were detected in June (46% loss for 8 mg prot/mL and 14% loss for 10 mg prot/mL), while October’s inhibition rates increased slightly (on average, 65.02% for 8 mg prot/mL and 62.58% for 10 mg prot/mL). A previous study on dried *P. palmata* determined an IC_50_ of water-soluble extracts of >1 kDa of 27.69 mg extract/mL, corresponding to 5.31 mg proteins/mL at 50% ACE inhibition [43]. Thus, the amount of protein and peptide required to inhibit ACE in the extract was slightly less than that required for the October extracts and considerably less than for the June extracts [43]. Finally, in this study, the effect of the drying process appeared to be strongly related to the harvest season and in favor of October. The loss of ACE inhibitory effect in June indicated an absence of protein degradation by drying, and as shown by the chemical composition of dried *P. palmata* (in Section 2.1.2) and with the PEY, it was probably not due to protein losses.

## 3. Materials and Methods

### 3.1. Macroalgal Biomass and Processing

Wild *P. palmata* (Linnaeus) Weber and Mohr were harvested in the Forillon area (QC, Canada) on 29 October 2019 and 2 June 2020. Collected specimens were cleaned with fresh water, and stipes and biofouling were removed. All samples were stored at 4 °C overnight. The next day, fresh *P. palmata* were divided into several batches to be either air dried for 5 h at 40 °C in a dryer (Hamilton beach, Glen Allen, VA, USA) then ground into 1 to 0.5 cm^2^ flakes or ground without pre-drying, as control batches. All samples were stored at −80 °C after grinding.

### 3.2. Chemical Composition (Lipids, Ash, Proteins, Carbohydrates, Fibers) and Mineral Composition

The same methods were applied as described in a previous study [41]. Briefly, seaweeds were freeze-dried and powdered (BFP660, Breville, Sydney, Australia). Moisture and ash contents were measured by a thermal gravimetric analysis using method no. 950.46 AOAC (Association of Official Analytical Chemists) 2008 and method no. 938.08 AOAC 2008, respectively. Protein content was obtained by the Kjeldahl method (AOAC 2008, no. 988.05) using a protein factor of 5 [87], and lipid content was obtained by the Bligh and Dyer method [88]. Total carbohydrate content was determined by difference [21], and fiber content was determined with a total dietary fiber assay kit (Megazyme, Bray, Ireland) using method no. 985.29 AOAC 2008. Calcium, iron, magnesium, potassium, and sodium contents were quantified using a flame atomic absorption spectrophotometer (220FS, Varian, CA, USA) with AOAC 2008 method no. 968.08, and iodine content was assessed with an iodine ion-selective electrode (Thermo Scientific 258508, Waltham, MA, USA).

### 3.3. Potential Bioactive Compound Determination

#### 3.3.1. Total Phenolic Content (TPC) and Carotenoids

The same methods were used as described in a previous study [41]. The TPC was determined using an adapted Folin–Ciocalteu method [89,90]. Briefly, aqueous extracts made with 0.5 g of freeze-dried/ground macroalgae were mixed with Folin–Ciocalteu reagent and the absorbance was read at 750 nm (Spectrafluor Plus, Tecan, Thermo Scientific, Ottawa, ON, Canada). The results were expressed as milligram Gallic acid equivalent (GAE)/g dry extract [91]. Carotenoid content was determined using Quan and Turner’s method [92] where 0.5 g of macroalgae sample was used for ethanol extraction and the absorbance of the extract was read at 468 nm (Genesys 20, Thermofisher, Waltham, MA, USA). The final concentration was calculated using the absorbance coefficient of astaxanthin and expressed as μg/g.

#### 3.3.2. Vitamins (β-Carotene and α-Tocopherol)

Vitamin quantification was performed following the protocol of Sanchez-Machado [93]. Dried macroalgae (0.25 g) was mixed with a solution of pyrocatechol (0.2 g/mL), and samples were saponified with 5 mL of KOH (0.5 M diluted in methanol) and heated at 80 °C in the dark. Vitamins were then extracted with 5 mL of hexane and transferred into a solution of chloroform and methanol (1:49, *v/v*). The separation was conducted by HPLC using a C18 Lichrospher (RP-18e 100 A, 125 × 4 mm, 5 µm) column coupled with a guard column, Security GuardTM C18 4 × 3.0 mm (Phenomenex, Torrance, CA, USA). The mobile phase was a solution of acetonitrile and methanol (70:30, *v/v*) heated to 25 °C, at a flow rate of 1 mL/min. The detection and quantification of β-carotene (450 nm) and α-tocopherol (298 nm) were carried out with a photodiode array detector (PAD) (ProStar 330, Varian, Palo Alto, CA, USA) and standards (α-tocopherol 258024, β-carotene 217538, Sigma-Aldrich, Saint-Louis, MO, USA).

### 3.4. In Vitro Bioactive Potential

#### 3.4.1. Water-Soluble *P. palmata* Extracts

The same methods were used as described in a previous study [41]. Briefly, samples were extracted twice with phosphate buffer (20 mM, pH 7) at a ratio of 1:1.665 (*w/v*) for non-dehydrated specimens and 1:16.667 (*w/v*) for dehydrated specimens. Samples were centrifuged at 4000× *g*, 4 °C for 45 min (Avanti^®^ J-E high-speed centrifuge, Beckman Coulter, Brea, CA, USA). Supernatants were collected, brought to 80% saturated ammonium sulfate, stirred, then centrifuged at 10,000× *g*, 4 °C for 60 min. Pellets were solubilized in Milli-Q water and dialyzed (1 kDa cut-off membrane unit, MWCO 1 kDa, Pur-A-Lyzer, Sigma-Aldrich, Saint-Louis, MO, USA) at 9 °C for 48 h. The resulting water-soluble seaweed extracts were lyophilized and stored in the dark in vacuum bags at −20 °C until further use. Total protein content was quantified according to the Dumas method [94], with a TruSpec N nitrogen analyzer (Leco Corporation, St. Joseph, MI, USA). All extractions were performed in triplicate.

#### 3.4.2. Oxygen Radical Absorbance Capacity (Orac) Assay

The methods used were the same as those described in a previous study [41]. Water-soluble seaweed extracts were dissolved in phosphate buffer (75 mM, pH 7.4) at eight concentrations (serial dilution from 5000 μg/mL to 39 μg/mL with a dilution factor of 2). Samples, or Trolox standards, or blanks (phosphate buffer) were put in black microplate wells (96-Well, Black U-Shape, Greiner Bio-One GmbH, Frickenhausen, Germany) with 150 μL of fluorescein (0.1 μM). The microplate was incubated and a volume of 50 μL of 2,2′-azobis (2-methylpropionamidine) dihydrochloride (AAPH) solution (150 nM) was added to each well. The fluorescence was read at a wavelength of 485 nm for excitation and 583 nm for emission and using Synergy H1 (Biotek, Winooski, VT, USA). The calculated ORAC values were expressed in μmol equivalent Trolox per gram of freeze-dried extracts (μmol TE·g^−1^). Assays were performed in triplicate.

#### 3.4.3. Ferric Ion Reducing Antioxidant Power (FRAP) Assay

The same methods were used as described in a previous study [41]. Macroalgae extracts (<1 kDa) were solubilized in Milli-Q water at five different concentrations (500, 250, 100, 50, and 10 μg/mL). A volume of 180 μL of FRAP reagent was added to the wells of a 96-well microplate (Greiner Bio-One, Frickenhausen, Germany) and incubated. Each tested concentration, Trolox standard, or blank (Milli-Q water) was added to the wells and incubated. The absorbance was read at 593 nm using a Microplate Absorbance Spectrophotometer (xMark, Biorad, Hercules, CA, USA). The calculated FRAP values were expressed as μmol equivalent Trolox per gram of freeze-dried extracts (μmol TE·g^−1^). Tests were performed in triplicate.

#### 3.4.4. Angiotensin-Converting Enzyme (Ace) Inhibition

The ACE-inhibitory activity was performed as described by Hayakari [95] and used by Hell et al. [43]. Briefly, 20 μL of ACE (250 mU per mL of borate buffer, 1 M NaCl, pH 8.3) and 80 μL of phosphate buffer (pH 8.3) were added to 20 μL of water-soluble macroalgae extract at concentrations of 8 and 10 mg prot/mL (*A_sample_*), or to enalapril which was used as the positive control (*A_control_*), then incubated at 37 °C for 10 min. Simultaneously, a replicate of each sample was incubated at 95 °C for 10 min to inactivate ACE and used as negative control (*A_blank_*). Then, 40 μL of Hippuryl-L-histidyl-L-leucine hydrate 6.25 mM (ACE substrate) was added to all samples and incubated for 60 min at 37 °C. The enzymatic reaction was stopped by heating to 95 °C for 10 min. Volumes of 360 μL of 2,4,6-trichloro-s-triazine (TT, 3% *v/v* in 1,4-dioxane) and 480 μL of phosphate buffer (pH 8.3) were added to each sample, vortexed, and centrifuged at 2000 ×*g* for 30 s (MiniStar, VWR, St. Catharines, ON, Canada). A transparent 96-well microplate (Clear F-Bottom, Greiner Bio-One GmbH, Frickenhausen, Germany) was filled with 200 μL of each sample per well and the absorbance was read at 382 nm (xMark, Biorad, Hercules, CA, USA). The ACE-inhibition rate (%) was calculated using the following formula:$$\mathrm{ACE}\;\mathrm{inhibition}\;(\%)=\frac{A_{control}-A_{sample}}{A_{control}-A_{blank}}\;\times \;100\;$$

### 3.5. Statistical Analysis

All laboratory analyses were performed in triplicate. All values are expressed as the mean ± standard deviation (SD). Statistical analyses were mostly performed using SAS University^®^ (Cary, NC, USA) using orthogonal contrasts. After checking for the normality and homoscedasticity of samples (using the Fisher and Shapiro–Wilk tests, respectively), ANOVA tests were performed with a threshold of α = 0.05, and the Tukey test was used as a post hoc test. For some cases where the ANOVA conditions were not reached, Prism GraphPad^®^ (San Diego, CA, USA) software was used and the Kruskal–Wallis test followed by a multiple comparison (Dunn’s test) was performed, also with a threshold of α = 0.05.

## 4. Conclusions

In conclusion, wild crude *P. palmata* showed an important seasonal variation in chemical, mineral, and bioactive compound composition, with markers of growth in June and decreased growth during October. Indeed, June specimens were richer in lipids, proteins, ash, fiber, sodium, calcium, magnesium, iron, polyphenols, α-tocopherol, and β-carotene, signaling an active metabolism. Conversely, macroalgae harvested in October were significantly richer in carbohydrates, which probably reflects the increased synthesis of small sugars called floridosides. October specimens were also richer in carotenoids, probably caused by a decrease in photoperiod and light intensity combined with an increase in nitrate concentrations in the water. ORAC, FRAP, and ACE inhibition activities measured for water-soluble extracts at >1 kDa, containing proteins and peptides, did not significantly differ through the harvesting months despite significant differences in the protein content of crude macroalgae. This indicates a weak impact of the season on the antioxidant and ACE-inhibitory potential of *P. palmata* from Eastern Quebec. For better nutritional value and health benefit potential, *P. palmata* might be harvested in June. However, *P. palmata* growth yield and shoreline abundance should be measured to better reflect the algal industry reality. In fact, some companies in Quebec harvest *P. palmata* mostly during the autumn because, in June, the available biomass in their area is too small. Since the fiber content was similar between the two months, harvesting *P. palmata* in October could be more profitable for the valorization of polysaccharides. Moreover, crude *P. palmata* harvested in both months presented higher amounts of magnesium than chocolate (70–85% of cocoa), more calcium than watercress, more iron than parsley, and a better Na/K ratio than shrimp, another marine product. The drying process induced only small changes for most analyzed compounds of *P. palmata*. By far, the factor determining the best nutrient potential was the harvest season. The drying process did not promote a large loss or gain of bioactive potential, thus indicating no potential negative impact during processing at a temperature of 40 °C, which limited the deleterious impact of the heat. Drying would therefore seem a good strategy for macroalgae quality and preservation.

## Figures and Tables

**Figure 1 marinedrugs-21-00392-f001:**
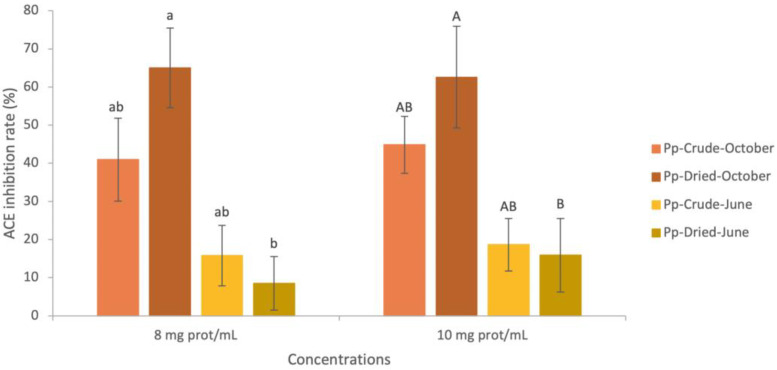
Angiotensin-converting enzyme (ACE) inhibition activity in water-soluble extracts from *Palmaria palmata* (crude and dried). Results are expressed as ACE-inhibition rate in percentages (%) of water-soluble extracts (mean ± SD; *n* = 3). Means with different lowercase letters (a,b) differ significantly (*p* < 0.05). Means with different capital letters (A,B) differ significantly (*p* < 0.05). No comparison was made between the different concentrations. Pp: *Palmaria palmata*.

**Table 1 marinedrugs-21-00392-t001:** Chemical composition of crude and processed wild *P. palmata* harvested in October 2019 and June 2020.

*P. palmata*	Lipids (%)	Proteins (%)	Ash (%)	Carbohydrates (%)	Fiber (%)
Crudes—October	1.17 ± 0.18 ^d^	11.23 ± 0.17 ^c^	12.30 ± 0.04 ^c^	75.30 ± 0.20 ^a^	35.81 ± 0.27 ^c^
Dried—October	1.65 ± 0.10 ^c^	11.19 ± 0.17 ^c^	12.22 ± 0.09 ^c^	74.94 ± 0.13 ^ab^	36.54 ± 0.18 ^b^
Crudes—June	4.64 ± 0.19 ^a^	16.19 ± 0.09 ^b^	27.26 ± 0.29 ^a^	51.91 ± 0.52 ^bc^	36.88 ± 0.10 ^b^
Dried—June	3.65 ± 0.21 ^b^	17.32 ± 0.17 ^a^	26.18 ± 0.05 ^b^	52.85 ± 0.43 ^ab^	37.80 ± 0.22 ^a^

Lipids, proteins, ash, carbohydrates, and fiber are expressed as a percentage of the DW of macroalgae (mean ± SD; *n* = 3). Means within each column with different letters (a–d) differ significantly (*p* < 0.05).

**Table 2 marinedrugs-21-00392-t002:** Mineral composition of crude and processed wild *P. palmata* harvested in October 2019 and June 2020.

*P. palmata*	I (mg/100 g)	K (g/100 g)	Na (g/100 g)	Ca (mg/100 g)	Mg (mg/100 g)	Fe (mg/100 g)	Na/K Ratio
Crudes—October	3.00 ± 0.44 ^bc^	3.43 ± 0.06 ^bc^	1.30 ± 0.00 ^c^	120.00 ± 13.00 ^c^	169.00 ± 4.00 ^c^	30.00 ± 1.00 ^c^	0.38
Dried—October	4.10 ± 0.10 ^ab^	3.53 ± 0.12 ^ab^	1.27 ± 0.06 ^c^	130.00 ± 13.00 ^c^	168.00 ± 5.00 ^c^	28.67 ± 0.56 ^c^	0.36
Crudes—June	9.60 ± 1.45 ^ab^	9.23 ± 0.38 ^ab^	2.58 ± 0.06 ^a^	317.00 ± 21.00 ^a^	313.00 ± 9.00 ^a^	436.67 ± 35.00 ^a^	0.28
Dried—June	10.73 ± 2.63 ^a^	12.65 ± 1.73 ^a^	2.20 ± 0.04 ^b^	256.00 ± 2.00 ^b^	261.00 ± 4.00 ^b^	352.33 ± 4.16 ^b^	0.17

Iodine (I), potassium (K), and sodium (Na) are expressed as g/100 g DW of seaweed, and calcium (Ca), magnesium (Mg), and iron (Fe) are expressed as mg/100 g (mean ± SD; *n* = 3). Means within each column with different letters (a–c) differ significantly (*p* < 0.05).

**Table 3 marinedrugs-21-00392-t003:** Bioactive compound contents of crude and processed wild *P. palmata* harvested in October 2019 and June 2020.

*P. palmata*	Carotenoids (μg/g)	Polyphenols (mg GAE/g)	α-Tocopherol (μg/g)	β-Carotene (μg/g)
Crudes—October	371.00 ± 11.00 ^b^	1.10 ± 0.00 ^c^	9.95 ± 1.11 ^c^	12.87 ± 3.09 ^c^
Dried—October	308.00 ± 3.00 ^c^	0.80 ± 0.14 ^c^	9.05 ± 0.63 ^c^	7.73 ± 0.77 ^c^
Crudes—June	315.00 ± 22.00 ^c^	2.06 ± 0.05 ^b^	22.67 ± 2.08 ^b^	33.33 ± 1.53 ^b^
Dried—June	535.00 ± 24.00 ^a^	1.89 ± 0.17 ^a^	50.00 ± 5.29 ^a^	47.67 ± 2.08 ^a^

Carotenoids, α-tocopherol, and β-carotene are expressed as μg/g and polyphenols as mg GA/g (mean ± SD; *n* = 3). Means within each column with different letters (a–c) differ significantly (*p* < 0.05).

**Table 4 marinedrugs-21-00392-t004:** Water-soluble extracts from wild and dried *P. palmata*: proportion of protein, extraction yield, and protein extraction yield.

*P. palmata*	Crudes—October	Dried—October	Crudes—June	Dried—June
Protein (%)	15.70 ± 2.31 ^a^	23.00 ± 0.89 ^ab^	29.03 ± 4.30 ^bc^	22.32 ± 2.34 ^ab^
Extraction yield (%)	8.40 ± 0.24 ^a^	3.44 ± 1.25 ^bc^	5.60 ± 0.47 ^ab^	5.38 ± 0.20 ^ab^
Protein extraction yield (%)	12.64 ± 1.57 ^a^	10.37 ± 1.28 ^a^	7.45 ± 1.16 ^a^	6.92 ± 0.62 ^a^

In the table, “Protein” represents the amount of protein that was obtained in each extract, “Extraction yield” represents the difference between the initial mass and the mass of the extract and “Protein Extraction Yield” represents the amount of protein in the extract as a function of the mass of the extract. Protein, extraction yield, and protein extraction yield are expressed as a percentage of the DW of seaweed (mean ± SD; *n* = 3). Means within each row with different letters (a–c) differ significantly (*p* < 0.05).

## Data Availability

The datasets analyzed in this study are available in a publicly accessible repository that can be found here: Merinov and Université du Québec à Rimouski (2022). Banque des données sur les macroalgues marines des sites de la Gaspésie (2019) [Data set]. https://catalogue.ogsl.ca/dataset/ca-cioos_cbf7730a-7074-4a37-92c8-0e549d6c6194?local=fr, accessed on 21 April 2023.

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
