# Peer review of "Impact of Harvest Month and Drying Process on the Nutritional and Bioactive Properties of Wild Palmaria palmata from Atlantic Canada"

_marinedrugs, 2023, doi:10.3390/md21070392_

Round 1
Reviewer 1 Report
Line 104 & 105: Do you mean application of low temperature drying or freezing? If you mean freezing could you change to low temperature storage to make it more clear?
From line 304-327 there are a lot of comparisons to other foods, are these foods known to be high in these minerals? This could be more easily visualised in a graph or table.
It would be good in the introduction to have a paragraph explaining some of the limitations with comparing a lot of these studies. If you could summarise the different methods for measuring antioxidant activity for instance, not only this but also the different extraction techniques for each assay make it difficult to compare studies.
Line 497 need to remove the word ‘would’ in the sentence
I think it would be nice for more of the results to be visualised graphically.
Why did you decide to use the methods that you used? Does ORAC for instance give advantage over DPPH? Are there more studies using this to compare to? FRAP is obviously a chelation reaction so it is nice that you have looked into two different mechanisms of antioxidant measurement but this does not come across in the paper.
The impact of this work needs to be highlighted in the conclusion; you state that carbohydrates are higher in the autumn is this not still an important area of nutrition? Also, if this is the case could it be fed into other industries such as sugar refinement which does not compete for land space. Also if the micronutrients are significantly higher in the summer months could you get away with a smaller harvest for the same yield of bio actives?
Author Response
Comments and Suggestions for Authors
- Line 104 & 105: Do you mean application of low temperature drying or freezing? If you mean freezing could you change to low temperature storage to make it more clear?
Comment: It was for freezing method, so “(freezing)” was added to “low temperature” for clarity.
- From lines 304-327 there are a lot of comparisons to other foods, are these foods known to be high in these minerals? This could be more easily visualised in a graph or table.
Comments: The foods used to make comparisons are foods known for their richness in minerals or vitamins.
- Lines 304-327 concern the bioactive compounds content section of crude palmata, and references were added to justify the food choice.
- Lines 233-255 concern the mineral composition of crude palmata and references were also added to justify the food choice.
Comparison with common foods of some potential minerals or bioactive compounds of P. palmata were pointed out and it was decided not to add new table or a graph.
- It would be good in the introduction to have a paragraph explaining some of the limitations with comparing a lot of these studies. If you could summarise the different methods for measuring antioxidant activity for instance, not only this but also the different extraction techniques for each assay make it difficult to compare studies.
Comment: A paragraph has been added in the introduction.
- Line 497 need to remove the word ‘would’ in the sentence
Comment: “would” has been removed.
- I think it would be nice for more of the results to be visualised graphically.
Comment: Table 5 (ACE inhibition rates) was changed for a Figure 3 (graph).
- Why did you decide to use the methods that you used? Does ORAC for instance give advantage over DPPH? Are there more studies using this to compare to? FRAP is obviously a chelation reaction so it is nice that you have looked into two different mechanisms of antioxidant measurement but this does not come across in the paper.
Comment: We performed preliminary tests at the beginning of the research. The DPPH test was highly variable on our P. palmata extracts and not accurate. Similarly, previous studies on P. palmata (such as Harnedy, P. A., Soler-Vila, A., Edwards, M. D., & FitzGerald, R. J. (2014). The effect of time and origin of harvest on the in vitro biological activity of Palmaria palmata protein hydrolysates. Food research international, 62, 746-752.) only assess antioxidant activity with ORAC and FRAP assays. We therefore decided to choose the 2 tests (ORAC and FRAP) instead of 3, since the use of a single technique is not recommended (Boisvert, C., Beaulieu, L., Bonnet, C., & Pelletier, É. (2015). Assessment of the antioxidant and antibacterial activities of three species of edible seaweeds. Journal of Food Biochemistry, 39(4), 377-387.).
The difference between ORAC and FRAP is presented in 2.4.2 (crude P. palmata section lines 458-461) and information about antioxidant methods and extractions was added in the introduction, as suggested.
- The impact of this work needs to be highlighted in the conclusion; you state that carbohydrates are higher in the autumn is this not still an important area of nutrition? Also, if this is the case could it be fed into other industries such as sugar refinement which does not compete for land space.
Comment: We considered June might have the best nutritional quality because the composition reflects a greater presence of the majority of nutrients.
As describe in the results/discussion section 2.1.1, P. palmata harvested in October contained more carbohydrates than those of June, but the fiber content was similar between the two months. This difference suggests that the carbohydrates produced in October could be simple sugars such as floridoside (the main C reserve for P. palmata - Martinez, B.; Rico, J.M. Changes in Nutrient Content of Palmaria Palmata in Response to Variable Light and Upwelling in Northern Spain(1). J Phycol 2008, 44, 50-59, doi:10.1111/j.1529-8817.2007.00440.x.). Since floridoside “plays a role, as does sucrose in higher plants” (Chen, Q., Yu, X., Liu, S., Luo, S., Chen, X., Xu, N., & Sun, X. (2022). Identification, Characteristics and Function of Phosphoglucomutase (PGM) in the Agar Biosynthesis and Carbon Flux in the Agarophyte Gracilariopsis lemaneiformis (Rhodophyta). Marine Drugs, 20(7), 442.) it could have an application in the food industry, but information about it lacks.
However, since fibers could have an interesting use in the food industry, we have added this information in the conclusion.
Also if the micronutrients are significantly higher in the summer months could you get away with a smaller harvest for the same yield of bio actives?
The bioactivities (antioxidant and ACE-inhibition) of P. palmata were measured using water-soluble extracts. During extraction, we mainly extracted proteins and peptides but also some water-soluble polyphenols and polysaccharides. Thus, the bioactivities presented reflect only the potential of extracted water-soluble molecules.
Since asthaxantin [carotenoids] (Bjørklund, G., Gasmi, A., Lenchyk, L., Shanaida, M., Zafar, S., Mujawdiya, P. K., ... & Peana, M. (2022). The Role of Astaxanthin as a Nutraceutical in Health and Age-Related Conditions. Molecules, 27(21), 7167.), β-carotene and ?-tocopherol (Webster, R. D. (2012). Voltammetry of the liposoluble vitamins (A, D, E and K) in organic solvents. The Chemical Record, 12(1), 188-200.) are liposoluble molecules, they are expected to be slightly found in P. palmata water-soluble extracts.
So, the harvesting of P. palmata in June, despite a lower biomass availability, should be chosen if bioactivity (through vitamins, antioxidant and ACE inhibition) is desired, but obviously it depends on the use as well as the necessary quantities.
In order to clarify, “water-soluble extracts” has been specified in the conclusion.
Reviewer 2 Report
The publication submitted for evaluation contains the results of research on Palmata palmaria from Quebec were harvested in October 2019 and June 2020, dried (40 °C, ≃ 5h) or stored as frozen controls (-80 °C). Comparing the content of active compounds in biomass collected from the same area over several months is not a bad idea - however, I am surprised that the authors did not choose one consistent growing season, e.g. June 2020 and October 2020, or June 2019 and October 2019 The authors have a number of works on similar topics to their credit, hence the research workshop and work methodology are well defined.
Author Response
Comments and Suggestions for Authors
The publication submitted for evaluation contains the results of research on Palmata palmaria from Quebec were harvested in October 2019 and June 2020, dried (40 °C, ≃ 5h) or stored as frozen controls (-80 °C). Comparing the content of active compounds in biomass collected from the same area over several months is not a bad idea - however, I am surprised that the authors did not choose one consistent growing season, e.g. June 2020 and October 2020, or June 2019 and October 2019 The authors have a number of works on similar topics to their credit, hence the research workshop and work methodology are well defined.
Comment: The choice of harvesting periods was necessarily based on the funding period of the study and the seasonal cycle. Harvesting in 2019 and 2020 was also a matter of logistics of the project.
Modifications have been made on the manuscript:
- A paragraph on antioxidant testing and extraction methods has been added to the introduction.
- The legend of the table 4 has been extended for more clarity.
- Table 5 has been changed for Figure 1.
Throughout the manuscript, the discussion has been enhanced.
Reviewer 3 Report
The paper provides an extensive characterisation of a macroalgal species named P. Palmata harvested at different periods of the year. It analyses the effect of drying on the composition of P. Palmata as well as investigating its bioactive potential. The study is well performed and appropriate techniques were used to obtain the desired answers. However, the study seems a bit simplistic and doesn't provide much added value to the seaweed technology. Nonetheless, the part that is valuable is the fact of characterising the seaweed that was harvested at different periods of the year. This part will help the grower find a better trade off between quantity and quality of the seaweed. Below are some suggestions that could be helpful to the authors:
- Make sure to write P. palmata in Italic all over the manuscript
- Regarding section 3.2, please provide more information on each analytical method (proteins, ash, lipids, carbohydrates, minerals and fibres). Simply relying on a reference is not enough.
- Section 3.1, what is the species code of P. palmata ?
- Please correct the typo in Line 497
- Please correct the typo in Line 605
- Sub-section 2.1 is not very needed because it is predictable to have the same primary composition regardless if the seaweed is dried or crude.
- Section 2.4.1, the approach to proteins is not very accurate. For instance, Lowry method usually takes into consideration soluble proteins whereas Dumas method takes into consideration the total nitrogen content, which explains the difference in protein yield. The effect of defrosting or drying is usually on the structural integrity of the cell wall and the intracellular organelles, which affects (positively or negatively) the efficiency of protein extraction.
- Some of the result and discussion sub-sections will need to be enriched with a better scientific discussion. Comparing the results obtained in the study with the literature is very good, but not enough for a scientific discussion. The authors should explain why - for instance - why the protein content was higher in June? is it because of more sun exposure? is it due to harvesting of the seaweed at a specific growth phase (lag phase, log phase, exponential phase, stationary phase)? The same reasoning should apply to other sections where a better scientific discussion will be needed.
- English level of the manuscript is very good, but I suggest that the authors double read to manuscript to correct some typos.
Author Response
Comments and Suggestions for Authors
The paper provides an extensive characterisation of a macroalgal species named P. Palmata harvested at different periods of the year. It analyses the effect of drying on the composition of P. Palmata as well as investigating its bioactive potential. The study is well performed and appropriate techniques were used to obtain the desired answers.
Comment: Thank you.
However, the study seems a bit simplistic and doesn't provide much added value to the seaweed technology. Nonetheless, the part that is valuable is the fact of characterising the seaweed that was harvested at different periods of the year. This part will help the grower find a better trade off between quantity and quality of the seaweed.
Comment: Most of the published studies on dried P. palmata did not assess the composition in such an exhaustive way with chemical, mineral content details, vitamins, bioactive potential compounds, antioxidant activity and anti-ACE activity on the same batch. Moreover, in Québec (Gulf of St-Lawrence) research on this species is quite new and as the attractiveness of this macroalgae is growing it is mandatory to understand the impact of the drying process on its composition.
Below are some suggestions that could be helpful to the authors:
- Make sure to write palmatain Italic all over the manuscript
Comment: P. palmata has been italicized all over the manuscript, even in the references section.
- Regarding section 3.2, please provide more information on each analytical method (proteins, ash, lipids, carbohydrates, minerals and fibers). Simply relying on a reference is not enough.
Comment: Some details were added to the section.
- Section 3.1, what is the species code of palmata?
Comment: The specie code “(Linnaeus) Weber & Mohr” has been added.
- Please correct the typo in line 497
Comment: the typo “would” has been removed.
- Please correct the typo in line 605
Comment: the typo “The same methods” has been removed.
- Sub-section 2.1 is not very needed because it is predictable to have the same primary composition regardless if the seaweed is dried or crude.
Comment: Since the section 2.1.2 on dried P. palmata presented significant variations detected in fibers, ash, proteins and lipids and between the two months of harvest and according to the other reviewers, this section seems appropriate for the authors.
Moreover, as presented in the introduction, drying could affect the nutritional composition of seaweed, as illustrated in different papers (Abdollahi, M., Axelsson, J., Carlsson, N. G., Nylund, G. M., Albers, E., & Undeland, I. (2019). Effect of stabilization method and freeze/thaw-aided precipitation on structural and functional properties of proteins recovered from brown seaweed (Saccharina latissima). Food Hydrocolloids, 96, 140-150. ; Neoh, Y. Y., Matanjun, P., & Lee, J. S. (2016). Comparative study of drying methods on chemical constituents of Malaysian red seaweed. Drying Technology, 34(14), 1745-1751. ; Sappati, P. K., Nayak, B., VanWalsum, G. P., & Mulrey, O. T. (2019). Combined effects of seasonal variation and drying methods on the physicochemical properties and antioxidant activity of sugar kelp (Saccharina latissima). Journal of Applied Phycology, 31, 1311-1332.)
- Section 2.4.1, the approach to proteins is not very accurate. For instance, Lowry method usually takes into consideration soluble proteins whereas Dumas method takes into consideration the total nitrogen content, which explains the difference in protein yield. The effect of defrosting or drying is usually on the structural integrity of the cell wall and the intracellular organelles, which affects (positively or negatively) the efficiency of protein extraction.
Comment: Since the Lowry method is an approximate method and previous tests by the team showed that the Dumas method gave similar results to the Kjeldahl method on seaweed, we decided to use the Dumas method. Moreover, the nitrogen-to-protein conversion factor applied (5) was calculated by taking into consideration non-protein nitrogenous material and inorganic nitrogen (Angell, A. R., Mata, L., de Nys, R., & Paul, N. A. (2016). The protein content of seaweeds: a universal nitrogen-to-protein conversion factor of five. Journal of Applied Phycology, 28, 511-524.). The point highlighted in the text is that Harnedy & al, 2014 found on their extracts lower protein content in July than in October, while we found a higher content in June than in October.
To clarify, changes have been made to the article.
- Some of the result and discussion subsections will need to be enriched with a better scientific discussion. Comparing the results obtained in the study with the literature is very good, but not enough for a scientific discussion. The authors should explain why - for instance - why the protein content was higher in June? is it because of more sun exposure? is it due to harvesting of the seaweed at a specific growth phase (lag phase, log phase, exponential phase, stationary phase)? The same reasoning should apply to other sections where a better scientific discussion will be needed.
Comment: Some discussion elements were added in sections 2.1.2, 2.2.1, 2.3.2 and 2.4.1
For the section 2.4.1 – Crude P. palmata: authors chose not to explain the reason why crude seaweeds contained more proteins. This element was discussed in the section 2.1.1. An additional legend has been added to table 4, to explain the difference between the elements expressed in the table.
- English level of the manuscript is very good, but I suggest that the authors double read the manuscript to correct some typos.
Comment: The manuscript was double read and typos were removed.
Round 2
Reviewer 3 Report
No comments
Author Response
There was no comments from the reviewer 3.